# Ecogeographic Drivers of the Spatial Spread of Highly Pathogenic Avian Influenza Outbreaks in Europe and the United States, 2016–Early 2022

**DOI:** 10.3390/ijerph20116030

**Published:** 2023-06-01

**Authors:** Jonathon D. Gass, Nichola J. Hill, Lambodhar Damodaran, Elena N. Naumova, Felicia B. Nutter, Jonathan A. Runstadler

**Affiliations:** 1Department of Infectious Disease and Global Health, Cummings School of Veterinary Medicine, Tufts University, North Grafton, MA 01536, USA; 2Department of Public Health and Community Medicine, Tufts University School of Medicine, Boston, MA 02111, USA; 3Department of Biology, University of Massachusetts, Boston, Boston, MA 02125, USA; 4Institute of Bioinformatics, University of Georgia, Athens, GA 30602, USA; 5Friedman School of Nutrition Science and Policy, Tufts University, Boston, MA 02155, USA

**Keywords:** Influenza A virus, outbreak, wild birds, phylodynamic-GLM, Europe, North America, virus diffusion, phylogeography, BEAST

## Abstract

H5Nx highly pathogenic avian influenza (HPAI) viruses of clade 2.3.4.4 have caused outbreaks in Europe among wild and domestic birds since 2016 and were introduced to North America via wild migratory birds in December 2021. We examined the spatiotemporal extent of HPAI viruses across continents and characterized ecological and environmental predictors of virus spread between geographic regions by constructing a Bayesian phylodynamic generalized linear model (phylodynamic-GLM). The findings demonstrate localized epidemics of H5Nx throughout Europe in the first several years of the epizootic, followed by a singular branching point where H5N1 viruses were introduced to North America, likely via stopover locations throughout the North Atlantic. Once in the United States (US), H5Nx viruses spread at a greater rate between US-based regions as compared to prior spread in Europe. We established that geographic proximity is a predictor of virus spread between regions, implying that intercontinental transport across the Atlantic Ocean is relatively rare. An increase in mean ambient temperature over time was predictive of reduced H5Nx virus spread, which may reflect the effect of climate change on declines in host species abundance, decreased persistence of the virus in the environment, or changes in migratory patterns due to ecological alterations. Our data provide new knowledge about the spread and directionality of H5Nx virus dispersal in Europe and the US during an actively evolving intercontinental outbreak, including predictors of virus movement between regions, which will contribute to surveillance and mitigation strategies as the outbreak unfolds, and in future instances of uncontained avian spread of HPAI viruses.

## 1. Introduction

Highly pathogenic avian influenza (HPAI) viruses of clade 2.3.4.4 (H5Nx) emerged in Southeast Asia in 2014 prior to spreading across much of Asia, Europe, North America, and Africa, causing frequent outbreaks and high rates of mortality in wild and domestic birds [1,2,3,4,5,6,7,8]. H5Nx viruses, which are descendants of A/Goose/Guangdong/1/1996(H5N1) (Gs/GD), first detected in 1996 in China, frequently reassort with other HPAI and locally endemic low pathogenic subtypes, resulting in a constellation of novel reassortant virus lineages that have been isolated from a wide range of avian species [1]. Following their emergence in Asia in 2014, H5Nx viruses of clade 2.3.4.4 migrated with wild aquatic birds across the Pacific into North America later in the same year, causing outbreaks in wild and domestic birds until mid-2015 [4,9]. Despite the substantial, though short-lived, spread of H5Nx (specifically H5N8, H5N1, and H5N2) in North America from this incursion event via the Pacific route, H5Nx viruses have continuously circulated throughout Eurasia from 2016 until present, with a gradual but steady northward trajectory of virus movement in Europe observed between 2016 and 2021 [10,11]. In December 2021, the first-ever documented incursion of HPAI via the Atlantic route into North America was detected in St. John’s, Newfoundland and Labrador, Canada, and subsequently caused significant mortality among wild birds throughout the United States and Canada [12]. This novel introduction of H5Nx viruses into North America via the Atlantic route raises questions about factors that govern virus movement and spread within this multi-continent outbreak system (through March 2022) and how changes to these factors over time may have facilitated the first documented interhemispheric introduction of H5Nx viruses from Europe to North America in late 2021. 

Environmental, ecological, and anthropogenic factors have been investigated as drivers of host and virus movement via Bayesian phylodynamics previously, including H9N2 in Asia, H5N1 in Egypt, multiple subtypes of Influenza A virus (IAV) among wildlife in North America, and Ebola virus during the 2013–2016 West African epidemic [13,14,15,16,17,18]. Northern temperate zones at the margins of the Atlantic Ocean are undergoing shifts in climate regimes (i.e., increased air temperature, precipitation, and sea-surface temperature) due to global climate change (GCC), which has been linked to alterations in avian host ecology including migratory patterns, reproduction cycles, and trophic interactions [19,20,21]. Northward shifts in population distributions, for example, may increase the density of susceptible hosts for IAV infection year-round and the risk for more frequent interhemispheric virus movement via short-distance flights across the Arctic perimeter [20,22,23,24]. Much remains unknown, however, about the impact of environmental factors on the dispersal of viral lineages, particularly during active epizootics of HPAI viruses at a multi-continental scale [21,25,26,27,28,29,30]. 

This study combines Bayesian phylodynamic and generalized linear modeling (Phylodynamic-GLM) to uncover ecological and environmental predictors of H5Nx HPAI virus diffusion in Europe and the US between 2016 and early 2022. We hypothesize that geographic proximity between regions and higher latitude are predictive of greater H5Nx HPAI virus movement. Additionally, we project that changes in air temperature and precipitation over time (which have almost universally increased on average across the globe in recent decades) will be predictive of decreased virus movement due to ecological disturbance these changes may cause, resulting in altered avian host distributions and lessened environmental virus survival due to higher air temperature and greater rainfall. Uncovering environmental and ecological factors that predict HPAI virus dispersal will provide increased insights into how current and future ecosystem shifts may impact host and pathogen ecology and will demonstrate the importance of climate-aware surveillance and mitigation strategies.

## 2. Materials and Methods

### 2.1. Dataset

All publicly available avian-derived (domestic and wild) H5Nx HA segment sequences of clade 2.3.4.4 from Europe and North America between 2016 and 2022 were downloaded from the Influenza Research Database (IRD) [31] on 12 May 2022 (n = 321). The date range for this analysis reflects the period of viral circulation in avian hosts in Europe and North America following its eradication from North America in 2016 and prior to subsequent introduction back to North America in 2021. We added 170 publicly available H5Nx HA sequences from 2021 to 2022, downloaded from GISAID on 15 May 2022, as these were unavailable on IRD at the time of sequence acquisition, and 15 novel H5N1 HA sequences from avian surveillance in Massachusetts, USA, by our research group in 2022 (described elsewhere [32]), totaling 546 HA sequences. Metadata for each sequence were collected, including sampling date, season, host species, and geographic sampling location. Only IAV sequences from wild avian species or environmental matrices were included. Duplicate sequences; sequences with less than 75% unambiguous bases; all vaccine derivative and recombinant sequences; and sequences with unavailable isolation date, location, or host species were excluded, resulting in 506 sequences. Downsampling was performed to ensure relative evenness of geographic state groupings while preserving genetic diversity of the dataset, using the geographic state and year for random stratification. To root and historically time-calibrate the tree, H5 subtype HA avian sequences from IRD were downloaded for the period 1979–2015 from Europe and North America and randomly downsampled by year, resulting in 33 historic sequences. These sequences were ‘masked’ to ensure their contribution to the tree structure but not to quantification of diffusion rates or the GLM [33]. The total downsampled dataset, including the outgroup (GISAID sequences from North America (n = 170), unpublished Massachusetts sequences acquired by our group (n = 15), publicly available H5Nx sequences from Europe 2016 to early 2022 (n = 160)), and historic sequences (n = 33) resulted in a total of 378 sequences (Appendix A). Multiple sequence alignments were performed using MUSCLE in Geneious Prime 2022.05.14 and trimmed to the open reading frame. 

### 2.2. Time-Scaled Bayesian Phylogenetic Analyses

Bayesian molecular clock analyses were conducted using the Markov chain Monte Carlo (MCMC) method in BEAST v.1.10.4 [34] to construct time-scaled phylogenetic trees. Phylogenetic analyses implemented a Generalized Time-Reversible model (GTR) of nucleotide substitution [35] with a gamma plus invariant sites distribution of site heterogeneity (the Yang96 model [36]), with a lognormal uncorrelated relaxed molecular clock [37], and a constant coalescent population model [38]. The BEAGLE library, which optimizes computational efficiency, was used [39]. Eight independent MCMC analyses were run for 200 million generations, sampled every 20,000 runs, and parameter convergence and effective sample size (ESS) (required to be >200) were evaluated in Tracer v.1.7.1 [40]. Using LogCombiner v.1.10.4, 10% or greater burn-in was removed from each run, and independent runs were combined to establish the maximum clade credibility (MCC) tree, from which the last 500 trees from the posterior distribution were extracted and used as the empirical tree set for all subsequent phylodynamic analyses [41]. Trees were visualized using Figtree v1.4.4 [42].

### 2.3. Discrete Trait Diffusion Analyses between Geographic Regions

To infer significant discrete trait transition rates along phylogenetic tree branches of H5Nx subtype HA sequences between geographic regions, discrete trait diffusion Bayesian phylodynamic analyses were performed using BEAST v.1.10.4 [34]. We used an asymmetric substitution model with Bayesian stochastic search variable selection (BSSVS) and a strict clock model to estimate the most parsimonious diffusion between discrete states [43]. Sampling locations were grouped into geographic state categories, based on modified National Oceanic and Atmospheric Administration (NOAA) historical climate regions (locations in the US) and grouped countries by latitude (locations in Europe). Posterior inference of the complete Markov jump history through time was evaluated by quantifying transitions between discrete states (Markov jumps, i.e., the frequency of transitions from one geographic state to another along phylogenetic branches) and the duration of time viruses spend in each discrete state (Markov rewards) [44]. 

### 2.4. Generalized Linear Models and Empirical Predictors

As an extension to the discrete trait diffusion models, GLMs were implemented to quantify and evaluate predictors of transitions between geographic states along phylogenetic branches. GLM models parameterize transition rates between discrete states as outcomes of a log-linear combination of matrixed covariate predictors [45]. Specifically, we modeled diffusion between geographic region states, using a non-reversible continuous-time Markov chain (CTMC) process expressed as an X-by-X rate matrix of discrete state change (Λ) among X discrete states (geographic regions) [45,46]. The rate of transition from discrete state *i* to discrete trait *j* (Λ*ij*) is modeled through a linearized log function which incorporates all pairwise predictors (*p*1, …, *pn*) in the following equation: logΛ*ij* = *β*1*δ*1 log(*p*1{*ij*}) + *β*2*δ*2 log(*p*2{*ij*}) + ⋯ + *βnδn* log(*pn*{*ij*}), 
where *βi* is the relative contribution of predictor pi to the whole GLM across the empirical phylogenetic tree space, and δ is a binary indicator of a predictor’s inclusion in the MCMC simulation [46,47]. A Bernoulli prior probability distribution was used to equally weigh the probability that a given predictor would be included or excluded from the model [45]. The probability that a single parameter is included in the model is as follows: *qq* = 1 − e^[ln(*p*)/*n*], 
where *p* is the probability of no coefficients are included (0.5 as default), and n is the number of total coefficients. Statistical support for diffusion among discrete states was determined by Bayes Factors (BFs): 3 ≤ BF < 20, 20 ≤ BF <150, and BF ≥ 150, denoting positive, strong, and very strong support, respectively [48]. BFs represent the odds of the posterior probability (*pp*) of a coefficient’s inclusion in the model over its prior probability (*qq*):*BF* = [*pp***/**(1 − *pp*)]**/**[*qq***/**(1 − *qq*)], 
where *pp* is calculated from BSSVS results, and *qq* is calculated based on the prior assumption that there is a 50% probability that none of the coefficients are included in the model. In addition to BFs, 95% Highest Posterior Density (HPD) credible intervals were derived for each predictor, and these provide information on the certainty of each parameter value. Posterior probabilities were calculated to demonstrate a predictor’s inclusion in the model (only predictors with BF ≥ 3 and posterior probability ≥ 0.25 were considered statistically supported for model inclusion), and GLM coefficients provide conditional effect sizes for each predictor. All non-binary (i.e., not labeled as 0,1) continuous predictors were log-transformed and standardized prior to implementing the model in BEAST v.1.10.4 [34]; therefore, a GLM coefficient of 1.0 can be interpreted as an increase of one transition per year for every one unit increase in the log-transformed predictor. Each GLM was performed with at least four independent MCMC runs, containing 200 million generations which were sampled and logged every 20,000 runs.

To inform the diffusion of H5Nx viruses between geographic regions, several environmental, ecological, and geographical predictors were selected. Predictors were selected following a review of the literature regarding ecological and environmental factors that have been or are hypothesized to be associated with the movement and spread of IAV and other pathogens by wildlife [15,16,18,49,50]. We also included predictors not previously evaluated, including predictor value change through time (i.e., change in precipitation across years (mm)). These were selected based on hypotheses associated with the relationship between climate change and alterations in host–pathogen ecology of IAVs [20,23,51,52]. Variables reflecting the relationship between regions (i.e., distance between centroids, and shared borders) were included in the model once, whereas all other predictors were included twice, to measure the directionality of rates of virus transitions between geographic regions. For example, the values of average precipitation at both the geographic region of origin and destination of viral transitions were included to determine whether precipitation at the region of origin or the region of destination was predictive of spatial movement of H5Nx between geographic states. Continuous variables represent the average value for each geographic region and timeframe. All predictors are further described in Table 1.

## 3. Results

### 3.1. Bayesian Phylogeography of the H5Nx HPAI Virus Clade 2.3.4.4 Outbreak in Europe and United States, 2016 to Early 2022

H5Nx viruses circulated in Europe for almost 5 years prior to a singular intercontinental introduction event of H5N1 from Central Europe to the US (most likely via Canada, as detailed in the Discussion), specifically to South Carolina in the Midwest and Mid-Atlantic region (BF = 76) (Figure 1 and Appendix A).

Prior to and following the intercontinental incursion of H5N1 to North America in late 2021, H5Nx viruses circulated within Europe, transitioning with the most intensity and highest statistical support from Northern to Central Europe (BF = 47,475). Central Europe served as a highly significant source region to both southern (BF = 2152) and northern (BF = 14) regions (Appendix A). The Southern European region acted as a sink of virus and not a source back to other European regions (Figure 2A,B and Appendix A). H5Nx viruses circulated in Northern Europe for the greatest duration of time between 2016 and 2022 (32.8% of this time period), as measured by Markov rewards, which represent the mean proportion of time that viruses circulate in each geographic region during an outbreak, followed by Central Europe (30.0%) and Southern Europe (10.2%). H5Nx viruses continuously circulated in Mainland Europe for 73.0% of the time between 2016 and February 2022 (Appendix A). 

Following the intercontinental introduction to North America, H5N1 viruses spread rapidly between geographic regions in the United States. The Midwest and Mid-Atlantic region served as a significant source of virus to both the Upper Midwest (BF = 858) and Northern Rockies and Plains (BF = 1526). Transitions along an east–west axis occurred, for example, between the Upper Midwest and the Northern Rockies and Plains (eastward movement (BF = 9), westward movement (BF = 12)), though to a lesser extent than overall transitions along a predominantly northward in direction south–north axis (Figure 2A and Appendix A). Between approximately December 2021 and March 2022, after viral introduction to North America via the Atlantic route, H5N1 viruses circulated for the greatest duration of time in the Midwest and Mid-Atlantic USA (32.8%), 26.4% of time Northeastern USA, 21.9% of time in the Northern Rockies and Plains, and 18.9% of time in the Upper Midwest region (Appendix A). Of all geographic viral transitions between US regions (represented by Markov jumps, which are defined as the percent of regional transitions between regional states along phylogenetic tree branches), 37.1%, 33.1%, and 11.8% migrated at high rates from the Midwest and Mid-Atlantic to the Northern Rockies and Plains, Upper Midwest, and Northeast, respectively (Figure 2A,B and Appendix A). All other transitions between US regions occurred at rates less than 7%. Between December 2021 and March 2022, H5N1 viruses circulated and spread within the United States, with no statistically supported transitions back to any European regions (Figure 2A,B and Appendix A).

### 3.2. Generalized Linear Model of Ecological and Environmental Predictors of H5Nx Diffusion in Europe and the US, 2016–2022

Among continuous (Table 2) and binary predictor variables used to inform the phylodynamic-GLM, summary statistics ranged widely between geographic regions, including distance between centroids (km), latitude at origin, European region as a destination, annual temperature change at origin (°C), annual temperature change at destination (°C), and location USA at origin. Given that the analysis was completed using all publicly available H5N1 HA sequences in the United States during an ongoing outbreak, downsampling the dataset to ensure relatively similar sequence counts by region was not possible. To account for this variability (sample size per region ranged from 33 to 80) sample size was included as a covariate predictor in the model to evaluate whether inclusion probabilities of other predictors were sensitive to the sample sizes in each geographic state (Table 2). The final model contained 19 predictor variables, six of which were demonstrated to be statistically supportive of inclusion within the model of viral spread between geographic regions.

The phylodynamic-GLM analyses found that geographic proximity (i.e., the distance between geographic centroids of each region) revealed very strong support for model inclusion (BF = 253.24), demonstrating that virus movement between regions was inversely associated with greater distance between regional geographic centroids (median coefficient = −2.13). Overall, viral transitions to European regions resulted in less onward viral movement (median coefficient = −2.27; BF = 9.43); however, when the origin location of viral transitions was in the US, significantly greater viral movement was found (median coefficient = 0.92; BF = 6.22), revealing increased viral spread in the United States as compared within Europe following the December 2021 introduction of H5Nx viruses to North America via wild bird migration. Contrary to our hypothesis, northward movement of viral diffusion was not statistically associated with increased geographic spread (BF = 2.07), and a higher latitude at the origin of viral transitions between regions was associated with less viral movement overall (median coefficient = −2.27; BF = 12.29). The results demonstrated that a location in Central Europe was the most statistically supported origin for the intercontinental transmission of H5N1 to North America in late 2021. Greater changes in air temperature over time at the origin (BF = 17.44) and destination (BF = 9.28) of viral transitions between regions were found to be associated with decreased H5Nx spread in Europe and the US. The additional 13 predictors included in the model failed to demonstrate statistical support (Figure 3 and Appendix A).

## 4. Discussion

The findings from this study provide novel data on the migration of clade 2.3.4.4 H5Nx HPAI viruses among geographic regions within and between Europe and the US during 2016 and early 2022, as well as geographic and environmental predictors of virus spread. Specifically, we found that greater spread was associated with virus migration originating in US regions and between geographically proximal regions, and virus migration was negatively associated with regional temperature change overall. Given the novel Atlantic route introduction and unprecedented geographic scope of the outbreak among wild and domestic birds, as well as terrestrial and marine mammals, these data provide important ecogeographic context regarding factors predictive of virus spread within the dual continent outbreak system [1,2,32,53,54,55].

The first recorded introduction of H5N1 viruses from Europe to North America, most likely via long-distance migratory birds, was detected following an unusual mortality event on an exhibition farm in Newfoundland and Labrador, Canada, in December 2021. This was the first detection of H5Nx HPAI viruses in North America since previous outbreaks in 2014–2015 spread by wild birds migrating across the Pacific route from East Asia [12]. Our Bayesian phylodynamic analysis of the HA segment of H5Nx HPAI viruses in Europe and the US establishes that a divergence event occurred sometime during 2020–2021, where North American viruses split from their most recent common ancestor with European lineage H5Nx viruses (Figure 1 and Appendix A). Though H5N1 viruses were not detected in Newfoundland and Labrador until the end of 2021, circulation across the Atlantic may have been facilitated by (a) the seasonal spring migration of Anseriformes (e.g., Eurasian Wigeon, Barnacle Geese, or Greylag Geese) from Mainland Europe to breeding grounds in Iceland or Greenland, and (b) the subsequent autumn migration of several gull species (e.g., Greater Black-Backed, Lesser Black-Backed, and Black-Headed Gulls) whose pelagic migratory patterns link these regions with Northeastern Canada [12,56,57]. Though a direct nonstop transatlantic incursion is possible, it is far more probable that interspecies transmission events on breeding grounds in the North Atlantic, particularly from adult to immune-naïve juvenile birds, enabled the necessary conditions for intercontinental spread [12,24,58]. 

Our phylogeographic analysis supports data demonstrating that the first detected introduction to the United States occurred in South Carolina in the Midwest and Mid-Atlantic region in late December 2021 [58]. H5N1 was first detected in this region in an American Wigeon and American Blue-Winged Teal (Appendix A). Though the ancestral and epidemiological relationship between H5N1 isolates detected in Canada and South Carolina remain unknown, it is most likely that the virus was introduced to late-season birds migrating southward from Canada along the Atlantic flyway in late 2021, which then seeded populations of birds migrating northward through the Carolinas in the subsequent spring, as is supported by our phylogeographic analysis depicted in Figure 2A.

Geographic proximity has previously been shown to be predictive of virus movement between global regions [15,16,18]. In our analysis, less distance between geographic regions was associated with increased viral diffusion, specifically 2.13 times more transitions per year for every one unit increase in the log-transformed predictor value. Our findings support observations that short-distance transmission drives global spread of H5Nx HPAI among avian populations and that long-distance trans-ocean or trans-continental virus movement by a single infected bird is a less frequent ecologic phenomenon. Gravity models, in which infectious-disease transmission is a function of population size and geographic distance, were first described for Influenza epidemics by Viboud et al. in 2006, and later adopted by Dudas et al. (2017) regarding the spread of Ebolavirus during the West African epidemic during 2013–2016 [18,59]. Early work conducted by members of our group also demonstrated strong seasonal synchronization of human influenza outbreaks in the continental US, accounting for seasonal human migration patterns [60,61,62]. Our finding that geographic proximity predicts the viral spread of H5Nx HPAI follows the same mechanism. Our findings demonstrate that highly synchronized local epidemics occurred throughout Mainland Europe during 2016–2021, which led to time-bound interactions between infected and susceptible wild birds in Central Europe, facilitating the gradual migration of H5Nx HPAI viruses across the North Atlantic to North America.

Our findings also reveal greater geographic viral spread from regions originating in the USA. Given our data also demonstrate no viral transmission events from the USA back to Europe following the December 2021 introduction event, these findings signify greater spread within the USA (in 2021–2022) as compared to Europe (in 2016–2022). Reasons for such rapid and uncontrolled spread within the US may include the (a) differential native species composition of North American birds; (b) high rate of susceptibility of diverse immune-naïve avian species to H5N1; and/or (c) geographic size of the continent, which may encompass the entire annual cycle for many bird species [4], increasing the likelihood of endemic local circulation. Additionally, previous research has demonstrated that HPAI introduction events to North America have increased the likelihood of onward viral transport to South America via north–south axis flyways, as was recently documented by our group [24,63].

There are roughly four times as many individual species of birds in North America as compared to Europe (2059 species versus 544 species) [64,65]. While evidence is lacking with respect to the relationship between avian species diversity and IAV spread, recent data have demonstrated that transmission relies on ecologically divergent bird hosts, and taxonomic diversity is associated with differences in H5N1-associated wild bird mortality between global regions [63,66]. Increased virus diffusion in the US may also relate to the relative immune naivety of avian species in the US versus Europe (or differences in susceptibility between Old and New World species), given that H5Nx HPAI viruses did not circulate in the North American region for approximately 6 years between 2015 and 2021 [1]. Comparatively, the endemicity of H5Nx viruses in Europe could have created a cycle of largely immune species due to frequent exposure during 2016–2022, lessening opportunities for viral spread compared to the novel introduction event to North America and subsequent rapid spread. Phylogeographic data from the US also indicate far more south-to-north virus migration than east–west migrations in either direction. While data on interactions between migratory flyways in North America have historically been contradictory, a recent study demonstrated that IAV dispersal within flyways was up to 13 times greater than between flyways, suggesting that the predominant gradient of diffusion of IAVs transpires along the north–south axis of within-flyway migration in North America, which our findings support [15,67].

Environmental factors and their average change over time have been shown to impact both host and pathogen ecology for a variety of wildlife species and infectious diseases [52,66,68,69]. The role of temperature fluctuation in the origin and destination regions of viral transitions, defined as the difference (in °C) in 2020 from the mean temperature recorded during the preceding century (1901–2000), was determined to be a predictor of decreased geographic viral spread of H5Nx HPAI viruses in GLM analyses. Though our model does not measure the directionality of temperature change itself, air temperature in all regions featured in the analysis increased between the latter time periods; therefore, our findings indicate that greater increased mean temperature values through time in both origin and destination regions of viral transitions are predictive of reduced H5Nx HPAI virus diffusion. There are several explanations for this finding. First, increased temperature has been associated with decreased survival of IAVs in environmental matrices (a major contributor to seasonal transmission dynamics), resulting in decreased indirect exposure among birds migrating to and from regions with increased average temperature change over time [19,69,70,71,72,73,74]. Second, climate-change-driven wetland loss may result in shifts in migratory strategy (timing and length of stay) to regions with more stable environmental trends in temperature change over time. Previous research, for instance, has proposed that climate change, including increases in temperature, will alter the distribution, migratory behavior, and residency of avian hosts to regions with milder temperature fluctuation [20,21], which we suggest may concentrate host abundance and shedding away from temperature-variable regions to regions experiencing less variability in temperature shifts. Finally, it should be noted that environmental factors are unlikely to impact host and pathogen ecology in isolation, and interactions between precipitation and air temperature (and other factors) may be heterogenous between global regions, thus adding complexity that our phylodynamic-GLM is not designed to measure [75]. More research is needed to understand the combined impacts of temperature change and other environmental trends on IAV movement between global regions and among wild and domestic avian hosts.

Multiple research avenues derived from the findings of this study are warranted. First, while our data present important findings regarding H5Nx HPAI diffusion in Europe and the US, we encourage comparison of these findings to similar analytical output from the 2014–2015 outbreak in North America which was seeded by virus migration from Asia via the Pacific route. We expect that differences in species diversity and the rate of transmission between North American regions will elucidate important ecological differences between outbreaks seeded by Asian-versus-European-origin migratory birds. Such a comparative analysis will provide novel insights into the ecology of HPAI outbreaks in North America and will contribute to future predictive models and surveillance strategies. Second, there is a need to innovate methods to facilitate the integration of publicly available IAV sequence data with ecological and environmental data associated with sampling dates and GPS locations. Such an effort would make predictive models of ecogeographic drivers of IAV transmission and movement more accessible and precise. For instance, metadata associated with publicly available sequences are most often limited to origin location of sequence isolate, host species, sampling date, and subtype. An algorithm (similar to GeoBoost2, a natural language processing pipeline for location extraction of molecular sequences [76]) may be designed to extract important metadata from location attributes of IAV sequences, including but not limited to the following environmental metadata: elevation, mean air temperature in sampling season, mean precipitation in sampling season, Normalized Difference Vegetation Index (NDVI) value on date of sampling, mean wild avian population density, and mean domestic avian farm density, among others. These data would be easily downloadable using GenBank accession IDs and may facilitate ecological and environmental analyses associated with IAV movement, interspecies transmission, geographic spread, and environmental persistence, among many other topics. 

### 4.1. Limitations

This study does have limitations. First, at the time of the analysis, the availability of published sequence data for the actively unfolding outbreak of H5Nx of clade 2.3.4.4 in North America following its 2021 introduction was limited due to non-uniform practices related to publishing virus sequence data, among other reasons. Second, due to the fact that avian surveillance is not systematic across space and time, regional groupings for this analysis were necessarily geographically vast, encompassing multiple countries (Europe) and states (USA). For Europe, regional groupings were determined based on relative latitude, given that environmental factors tend to trend by the earth’s latitudinal gradient [77]. For the USA, regional groupings were determined by modifying NOAA’s US Climate Regions to ensure that all publicly accessible data from the USA would be included in the model and maintain a reasonable sample size, while preserving climactic consistency. USA regions of Northeast, Upper Midwest, and Northern Rockies and Plains were well-represented in terms of sample size and geographic coverage; however, the Midwest and Mid-Atlantic zone includes states within three distinct climate zones. While these states generally lie within the same latitudinal range, the Midwest and Mid-Atlantic regional grouping may have limited the model’s power regarding transitions involving this region or the GLM’s predictors themselves. Third, host immune defenses exert more selective pressures on the HA gene than other non-external genes; therefore, unmeasured non-ecological forces may have influenced our evolutionary phylogenetic reconstruction more so than if we had chosen to model one or more different gene segments. Fourth, Markov rewards, the duration of time viruses will circulate in each region relative to all global regions, indicated that H5Nx HPAI viruses circulated in Europe for 73% of the time in the sampling timeframe of 2016–March 2022. Inversely, during 27% of the latter period, viruses were circulating in the United States. While H5N1 was first detected in December 2021, these viruses could have been introduced prior to this date; however, it is highly doubtful that the introduction occurred approximately 1.6 years prior (the equivalent of 27% of time during 2016–March 2022). We believe that the inferred underestimation of duration of virus circulation in Europe is due to our downsampling strategy, which attempted to preserve virus diversity while ensuring relative evenness of sequence counts per region. The vast majority of viruses circulating during this time period was doing so in Europe; therefore, we could have downsampled the dataset relative to prevalence by region. However, this was not possible given the dearth of available prevalence data by region in the emergent North American outbreak at the time of analysis, which was restricted to several states within the US. Fifth, given the ongoing nature of the outbreak, data included and findings from this study are limited to a cross-section of time from January 2016 to March 2022. As the outbreak has evolved beyond March 2022, new data have emerged that warrant a follow-on inquiry, including H5N1 incursions via the Pacific route at about the same time as viruses were introduced across the Atlantic to North America in early 2022 [78]. Viral genetic sequences demonstrating H5N1 incursion events via the Pacific were not publicly available at the time of this analysis and may have contributed to a more comprehensive picture of viral diffusion in North America as of late 2021 until March 2022.

### 4.2. Conclusions

The findings from this study reveal the geographic extent and directionality of the H5Nx HPAI virus outbreak within and between Europe and the US from 2016 and throughout the early few months following the introduction to North America in 2022. The data demonstrate localized epidemics of H5Nx throughout Europe in the first several years of the epizootic, followed by a singular branching point where H5N1 viruses were introduced to North America across the Atlantic via wild migratory birds. Once in the US, H5Nx HPAI viruses spread at a greater rate between US-based regions along migratory flyways, and no evidence points to spread back to any European region by March 2022. Overall, our GLM demonstrated that geographic proximity is a predictor of virus diffusion between regions, which implies that intercontinental spread across the Atlantic is relatively rare and may coincide with spring migration of susceptible avian species to regions in the North Atlantic. Finally, greater positive temperature change in both origins and destinations of viral transitions was predictive of reduced H5Nx HPAI virus spread, which may reflect reduced environmental persistence of the virus in higher ambient temperatures, declines in host species abundance in regions with elevated temperature, and changes in migratory patterns due to ecological alterations. Our data provide new knowledge about the spread and directionality of H5Nx HPAI virus dispersal throughout Europe and the US, including predictors of virus movement between regions which will contribute to surveillance and mitigation strategies as the outbreak unfolds, and can be used for future instances of uncontained avian spread of HPAI viruses.

## Figures and Tables

**Figure 1 ijerph-20-06030-f001:**
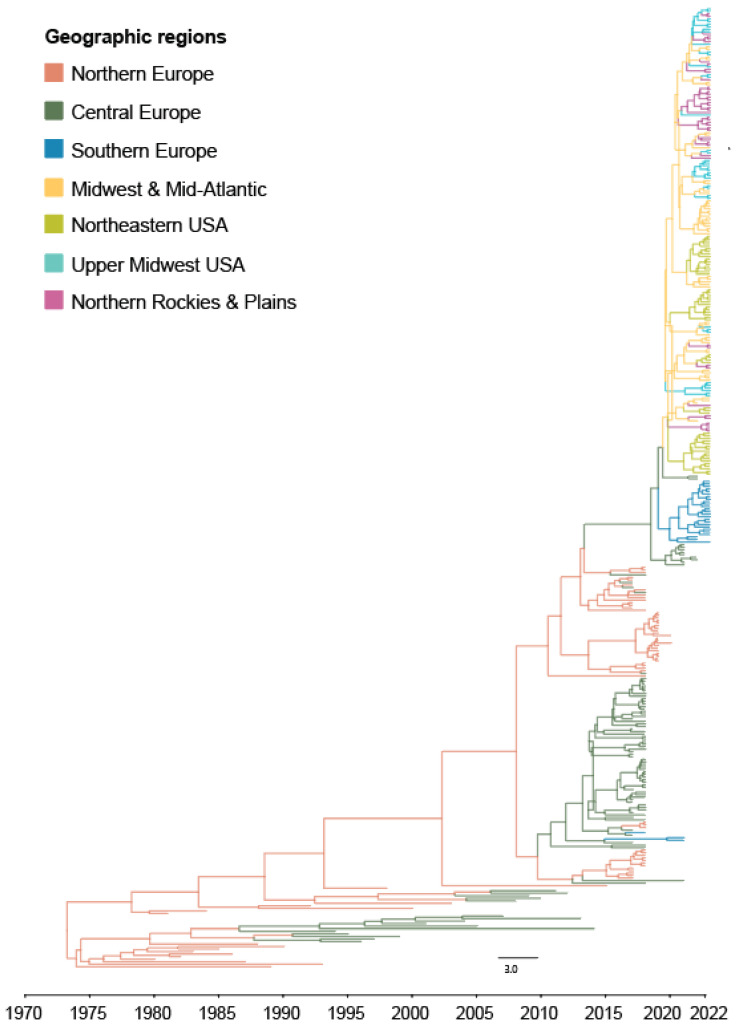
Markov Chain Monte Carlo (MCC) time-scaled phylogeographic tree of H5Nx Influenza A viruses (IAV) of clade 2.3.4.4. HA gene segments, color-coded by geographic source region. The phylogeographic tree with 95% Highest Posterior Density intervals featured in Appendix A.

**Figure 2 ijerph-20-06030-f002:**
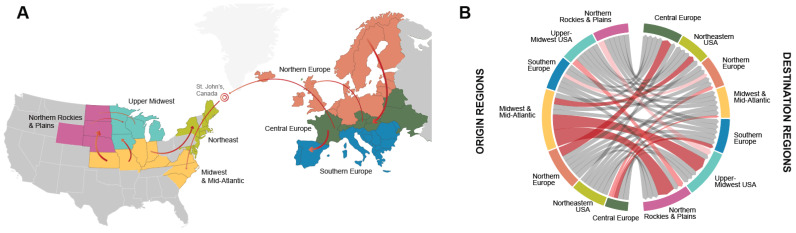
(**A**) H5Nx circulation in Europe and the US during outbreaks between 2016 and 2022. Significant discrete phylogeographic transitions between regions are represented by arrows from Northern Europe to Central Europe (BF = 47575), Central Europe to Southern Europe (BF = 2152), Midwest and Mid-Atlantic to Northern Rockies and Plains (BF = 1526), Midwest and Mid-Atlantic to Upper Midwest (BF = 858), Midwest and Mid-Atlantic to Northeast USA (BF = 692), Central Europe to Midwest and Mid-Atlantic (BF = 76), Central Europe to Northern Europe (BF = 14), Upper Midwest USA to Northern Rockies and Plains (BF = 12), and Northern Rockies and Plains to Upper Midwest USA (BF = 9). Only BFs > 3 with corresponding posterior probability (PP) estimates > 0.25 are presented as statistically supported. Arrows signify directionality, and greater arrow width corresponds to higher BF support of phylogeographic transitions between intracontinental geographic states. The bullseye signifies the first detection of HPAI in December 2021 in St. John’s, Canada, likely via Iceland and/or Greenland (Caliendo et al., see references), prior to introduction to the Midwest and Mid-Atlantic region. BFs and PPs for state transitions between all regions can be found in Appendix A. Map is not drawn to scale. (**B**) Mean transition rates between global regions. Chord diagrams depicting Bayes factors (BFs) for virus movement between regions within and between Europe and the US. Chord width is proportional to the mean transition rate from origin regions to destination regions. Statistically supported BFs (BF > 3.0) are depicted by red-shaded chords, with the strength of statistical support increasing with intensity of shading from pink (statistically supported) to bright red (very strong statistical support). Gray chords demonstrate possible transition between regions that demonstrated no statistical support (Appendix A).

**Figure 3 ijerph-20-06030-f003:**
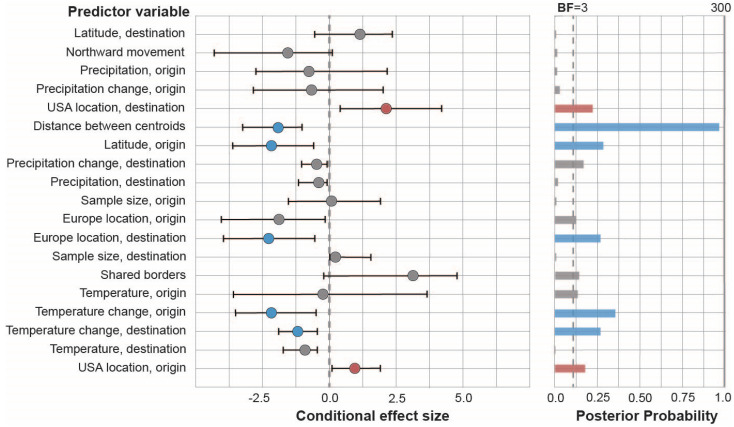
Predictors of migration of H5Nx HPAI viruses within and between Europe and US regions during outbreaks, 2016–2022. On the left panel, conditional effect size (circles) measures a predictor’s effect on the rate of migration, and 95% Highest Posterior Density credible intervals (black bars through circles) demonstrate certainty of the conditional effect. On the right panel, a predictor is included in the model when a Bayes factor (BF) > 3.0 (dashed vertical line) and posterior probability ≥ 0.20. Gray, blue, and red circles and bars signify no association, statistically supported negative association, and statistically supported positive association, respectively.

**Table 1 ijerph-20-06030-t001:** Environmental, ecological, and geographic predictors of virus diffusion in the Europe-North American system of H5Nx of clade 2.3.4.4.

Predictor	Justification	Value	Data Source
Distance between centroids	Decreased distance between states has been shown to relate to viral spread between geographic states in several phylogeographic-GLM models [15,16,17,45].	Great circle distance in kilometers (km) between geographic state centroids	Google Earth
Latitude	Given that many migratory avian species breed in northern latitudes, higher latitudes may increase global dispersal of IAVs due to transmission dynamics between adults and juveniles at breeding ranges, spreading viruses globally [13,15,16,17,24].	Decimal latitude at exact centroid of geographic region	Google Earth
Shared borders	Shared borders have been implicated in the spread of viruses between both humans and animals due to geographic proximity and movement patterns [13,50].	Binary 0/1 (no/yes)	Google Earth
Northward movement	Northward movement has been associated with global spread of IAVs due to the condensed land masses around the circumpolar perimeter of the Arctic that connect hemispheres, particularly following breeding season [10,11]. Northward movement has not been previously used as a predictor in standard GLMs.	Binary 0/1 (no/yes) whether geographic state of origin is north by latitude of state of destination	Google Earth
Precipitation	Precipitation has been modeled in GLMs previously, with varying significance [13,15,16,17,49].	Mean yearly precipitation in mm, 2020	US: NOAA National Centers for Environmental Information; Europe: The World Bank Open Data
Change in precipitation	Change in precipitation has not been previously used as a predictor in standard GLMs.	Difference between mean precipitation of 1901–2000 and that of 2020	US: NOAA National Centers for Environmental Information; Europe: The World Bank Open Data, The World Bank Group Climate Change Knowledge Portal
Air temperature	Air temperature has been modeled in GLMs previously, with varying significance [13,15,16,18,49].	Mean yearly air temperature in Celsius, 2020	US: NOAA National Centers for Environmental Information; Europe: The World Bank Open Data
Change in air temperature	Change in air temperature has not been previously used as a predictor in standard GLMs.	Difference between mean temperature of 1901–2000 and that of 2020	US: NOAA National Centers for Environmental Information; Europe: The World Bank Open Data, The World Bank Group Climate Change Knowledge Portal
Sample size	Commonly included in GLMs to account for different sample sizes by state, which can bias results. GLM assumes that the sample sizes across subpopulations are proportional to the subpopulation sizes [13].	Count of virus sequences from each geographic state	Downsampled dataset
Continental location of sequence: Europe or North America	There may be geographic differences in virus diffusion due to continental land size, proximity to nearby regions, or composition of host diversity and abundance in given regions. This has not been modeled previously.	Binary (0/1)	Downsampled dataset metadata

**Table 2 ijerph-20-06030-t002:** Summary statistics of continuous predictor variables used to inform the Bayesian discrete diffusion generalized linear model describing H5Nx HPAI of clade 2.3.4.4 diffusion in Europe and the US, from 2016 to early 2022. Additional binary predictors not featured in the table include shared borders, continent location USA (origin and destination), continent location Europe (origin and destination), and northward movement.

Variable Name	Mean	Standard Deviation	Range
Distance between centroids (km)	4587.3	3101.5	690–7564
Precipitation (mm)	838.2	262.8	473.8–1163.2
Precipitation change (mm)	34.8	38.2	−2.35–108.0
Temperature (°C)	10.13	2.66	7.60–14.39
Temperature change (°C)	1.37	0.092	1.23–1.48
Latitude	43.7	5.9	34.9–52.9
Sample size (count)	50.3	15.78	33–80

## Data Availability

All data supporting the findings of this study are openly accessible. Genetic sequences from IAV surveillance in North America and Europe were deposited in GenBank (https://www.ncbi.nlm.nih.gov/genbank) under the accession numbers listed in Appendix A. The raw data used for the analysis were deposited in the Dryad Digital Repository (https://doi.org/10.5061/dryad.4qrfj6qdk).

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
