# Peer review of "Ecogeographic Drivers of the Spatial Spread of Highly Pathogenic Avian Influenza Outbreaks in Europe and the United States, 2016–Early 2022"

_ijerph, 2023, doi:10.3390/ijerph20116030_

Round 1
Reviewer 1 Report
Gass: Ecogeographic drives of the spatial spread of HPAI outbreaks in Europe and North America, 2016-2022
The authors use Bayesian phylogenetics to attempt to gain further insight into viral movement within and between Europe and North America.
I wonder why the authors use 2016 as a starting point? 2014 was rather significant and comprises the a major viral incursion into north American that should be considered a part of this story. If there is a reason to start at 2016, this needs to be explained.
That the authors use data from only the lower 48 of the USA, and discuss dynamics in “North America” is rather bad form. There are other countries in North America beyond the USA which have also been struggling with HPAI outbreaks.
The authors indicate that only a single viral incursion occurred into North America, but the literature provides evidence for 2 independent incursions. This suggest not all sequences have been included, or that there are issues with the analysis. This should be resolved.
A threat from both sides: Multiple introductions of genetically distinct H5 HPAI viruses into Canada via both East Asia-Australasia/Pacific and Atlantic flyways. Virus Evolution Doi: https://doi.org/10.1093/ve/veac077
This study seems to ignore all sequences from North Africa and Asia. A recent study has demonstrated the key role of Egypt in the emergence of the 2.3.4.4b H5N1 lineage causing outbreaks today, and there has been substantial connectivity between Asia and Europe between 2016-2022. Without these sequences, any inference about patterns of virus movement in Europe are unfounded as key processes are ignored.
Key reference include:
The episodic resurgence of highly pathogenic avian influenza H5 virus. bioRxiv. doi: https://doi.org/10.1101/2022.12.18.520670
Bidirectional Movement of Emerging H5N8 Avian Influenza Viruses Between Europe and Asia via Migratory Birds Since Early 2020. Molecular biology and evolution. https://doi.org/10.1093/molbev/msad019
Minor comments
Line 42. “Gs/gd reassort with other HPAI”. Which HPAI? H9N2 and H7N9 are LPAI.
Line 45. “their emergence in asia in 2014, H5Nx”. Perhaps add “2.3.4.4.” here.
Line 52. St. John is in New Brunswick. I think the authors mean St. John’s, Newfoundland and Labrador.
Line 75, “Europe and north America”. The authors should change this to “Europe and the USA”. Although they don’t included Alaska.. so perhaps something even more restricted.
Line 100. How was downsampling performed. Manually? Random?
Line 104. Why 33 sequences? How did you choose? Only gs/gd or equal numbers in LPAI EU and LPAI N.Am?
Line 111. How did you deal with the gap in the cleavage site? MUSCLE isn’t great for gaps. Did oyu manually check?
Line 147-166. This is beyond my BEAST skills, so I leave it to the other reviewers to address the methods here.
Line 211-217. What I do know about trait analysis is that Markov jumps/rewards are highly sensitive to biases in data (if you have lots of sequences from one place relative to another, or temporal gaps). How was this addressed?
Line 261-263. Do the virus movement align with the direction of bird movement in the same time period?
Line 296-300. Am curious about the time element here and whether the data aligns with the direction of bird movement.
Reviewer 2 Report
Dear Editor,
The manuscript, entitled “Ecogeographic drivers of the spatial spread of highly pathogenic avian influenza outbreaks in Europe and North America, 2016-early 2022”, had examined the spatiotemporal extent of HPAI H5Nx viruses across continents and characterize ecological and environmental predictors of virus spread between geographic regions. The HPAI H5 viruses had introduced to North America twice in 2014 and 2021. This manuscript have investigated the later. The description of dataset is not clear, such as which viruses belonged to clade 2.3.4.4, what dataset was used to do GLMs analyses, and why ecogeographic drivers in 2020 were used instead of 2021 when the inter-continental transmission occurred.
Comments,
In Table 1, reference 48 should not be cited.
Figure 1, which viruses belonged to clade 2.3.4.4? why non A/Goose/Guangdong/1/1996(H5N1) lineage sequences were included? If they were also included in discrete trait diffusion analyses between geographic regions? The time scale was misplacement comparing with Supplementary figure S1.
Line 218-222, how the XX.X% of this time period in XXX region was calculated? In some period, the virus’ activity might very low to be detected, but they were continuing circulated in birds or they would be terminated. And the virus also can circulate and cause outbreak in multiple regions simultaneously.
Whether the ecogeographic drivers only contribute to H5Nx 2021 spread or also contribute to 2014 spread?
